# Parental hesitancy on COVID-19 vaccination of children under the age of 16: A cross-sectional mixed-methods study among factory workers

**Kyaw Thiha Aung[1], Ye Minn Htun[ID][1]*, Zin Lin Htet[1], Yan Naing Myint Soe[1], Phyo Ko Ko[2], Win Oo[2], May Soe Aung[ID][3], Tun Tun Win[2]**

1 Department of Prevention and Research Development of Hepatitis, AIDS and Other Viral Diseases, Health and Disease Control Unit, Nay Pyi Taw, Myanmar, 2 Department of Preventive and Social Medicine, Defence Services Medical Academy, Yangon, Myanmar, 3 Department of Preventive and Social Medicine, University of Medicine (1), Yangon, Myanmar

* dryeminnhtun85@gmail.com

## Abstract

### Background

Thanks to the development of COVID-19 vaccines, now they can be safely and effectively used to guard COVID-19 patients against severe illness, hospitalization, and even mortality. However, parents' unwillingness to vaccinate their children depends on a large extent on factors beyond the availability of vaccines, and understanding the factors associated with parental vaccine hesitancy has become increasingly important to the development of the COVID-19 vaccine program. Therefore, this study aimed to determine the parental COVID-19 vaccine hesitancy to their children and its associated factors among factory workers in Myanmar.

### Methods

A cross-sectional mixed-methods study was conducted as an explanatory sequential design, at Tri Star tyre factory (Ywar Ma), Yangon, Myanmar from August 2022 to February 2023. A total of 170 factory workers with children under the age of 16 participated in this study. The quantitative data were collected by the face-to-face interviews using a pretested structured questionnaire that included the Oxford COVID-19 vaccine hesitancy scale. Data were analyzed by using binary logistic regression to identify associated factors of parental hesitancy. Adjusted odds ratio (AOR) with a 95% confidence interval (CI) was computed to determine the level of significance with a p value ≤ 0.05. A subsample of 6 participants from each "hesitant group" and "non-hesitant group" towards COVID-19 vaccination was interviewed by the individual in-depth-interview guide to provide the reasons for their willingness or unwillingness to vaccinate to their children. The thematic analysis was undertaken for the qualitative data.

**Data availability statement:** All relevant data are within the manuscript and its Supporting Information files.

**Funding:** The author(s) received no specific funding for this work.

**Competing interests:** The authors have declared that no competing interests exist.

## Results

Among the total, 18.2% (95% CI: 12.7–24.9%) of the parents were hesitant to vaccinate their children against COVID-19 while 25.9% (95% CI: 19.5–33.1%) responded as unsure and 55.9% (95% CI: 48.1–63.5%) were non-hesitant for vaccination to their children. Male (AOR: 3.04, 95% CI: 1.35–6.84) and those who were not infected with SARS-CoV-2 (AOR: 2.66, 95% CI: 1.06–6.70) were significantly associated with parental COVID-19 vaccine hesitancy. The most common reasons for the unwillingness to receive the COVID-19 vaccination to their children were too young for vaccination, concerns about the safety of the vaccines, uncertainty about the effectiveness of the vaccines, and lack of trust in the origin of the vaccines.

## Conclusions

In this study, nearly one-fifth of the parents were hesitant to vaccinate their children against COVID-19. The findings of this study suggested that the government and healthcare professionals should provide health education about the importance of COVID-19 vaccination and the safety and efficacy of currently providing COVID-19 vaccines using mainstream media to improve the proportion of children getting vaccinated against COVID-19.

## Introduction

An outbreak of Coronavirus Disease 2019 (COVID-19) caused by a novel virus, the Severe Acute Respiratory Syndrome Coronavirus 2 (SARS-CoV-2), began in December 2019 in Wuhan, the capital city of the Hubei Province of China. After increasing the number of confirmed cases in China with the virus spreading rapidly to other countries, the World Health Organization (WHO) declared the viral epidemic a public health emergency of international concern on the 30th January 2020. Then WHO assessed that COVID-19 could be characterized as a pandemic on 11th March 2020, deeply concerned both by the alarming levels of spread and severity by the alarming levels of inaction [1]. COVID-19 is continuing to spread around the world with more than 758 million confirmed cases and more than 6.8 million deaths, at the end of February 2023 [2,3].

SARS-CoV-2 changes over time and these changes have little to no impact on the properties of the virus. However, some changes may affect the virus's properties, such as how easily it spreads, the associated disease severity, or the performance of vaccines, therapeutic medicines, diagnostic tools, or other public health and social measures [4]. Particularly in the event of the emergence of new SARS-CoV-2 variants, the non-pharmaceutical interventions are public health measures that aim to interrupt or reduce SARS-CoV-2 transmission in the community. The community mitigation strategies, ranging from individual actions such as regularly practicing good hand hygiene to more restrictive measures like limiting public gatherings, are implemented in combination and applied simultaneously [5,6]. While effective in controlling

the epidemic, some of these measures have significant socioeconomic costs and may negatively impact the physical and emotional well-being of populations [7].

In addition to this, vaccines reduce the risks of getting a disease by engaging with body's natural defenses of the body to build resistance to SARS-CoV-2 infections and make the immune system stronger [8]. Novel vaccines are constantly being developed to activate an immune response for the robust targeting of SARS-CoV-2 and its associated variants [9]. Safe and effective vaccines are currently available that provide strong protection against serious illness, hospitalization, and death from COVID-19 [10]. WHO recommended that the countries should continue to accomplish towards vaccinating at least 70% of their total populations, prioritizing the vaccination of 100% of healthcare workers and 100% of the most vulnerable groups, including people who are over 60 years of age and those who are immunocompromised or have underlying health conditions [11,12]. As of 28th February 2023, 69.6 per hundred people received at least one dose of COVID-19 vaccine globally [13].

Vaccine hesitancy, which expresses the unwillingness to get vaccines when vaccination services are available and accessible, is one of the greatest threats to global health. Whereas vaccine hesitancy has existed among a small percentage of people, its harmful effects are likely to be more pronounced during the COVID-19 pandemic. Some people are hesitant toward the COVID-19 vaccine due to its rapid development and distribution and those concerned about the safety of a vaccine developed much quicker than usual. COVID-19 vaccine hesitancy can produce substantial risks not only for people who delay or refuse to be vaccinated but also for the broader community. It will also make communities unable to reach thresholds of coverage necessary for herd immunity against COVID-19, thus unnecessarily continuing the pandemic and resulting in untold suffering and deaths [14,15].

Factory workers are essential for driving the production growth and play a crucial role in generating social wealth [16]. The effective COVID-19 vaccination program with the strategies of sustainable vaccination process among factory workers is vital for the recovery of economic sector [17]. Regarding the occupational status, frontline workers such as healthcare workers and workers in customer-facing industries were at higher risk of infection, and therefore these occupations should receive high priority for vaccination [18,19]. However, the vaccination coverage among industrial workers was usually lower than that of the general population [16]. Based on the findings of a previous study, antivaccine attitude, vaccine origin, education, marital status, having children, participants with chronic diseases were determinants of COVID-19 vaccine hesitancy among industrial workers [20].

A timely understanding of community responses to the COVID-19 vaccines is important for policy-making and service planning. For children under the age of 18 years, parents are usually the decision-makers regarding their children's vaccination. Hence, it is important to understand parents' COVID-19 vaccine hesitancy towards their children and its related factors [21]. As per prior studies, the parental vaccine hesitancy to COVID-19 is ranging from 15.1% to 61.9% around the world. Understanding the differences in parental COVID-19 vaccine hesitancy to their children across varying communities and sociodemographic groups plays a critical role in diminishing the burden of the pandemic through administering vaccinations as a preventive behaviour [22–28].

In the context of Myanmar, the first cases of COVID-19 were reported on 23rd March 2020 and there were 633,918 confirmed cases of COVID-19, including 19,490 deaths as of 28th February 2023 [2,29]. The National Deployment and Vaccination Plan against COVID-19 was launched on January 27, 2021, and is being implemented with the aims of decreasing COVID-19-related morbidity and mortality, limiting the transmission of diseases, and minimizing the economic burden of pandemic on the nation [29,30]. In June 2022, 64.6 per hundred people received at least one dose of COVID-19 vaccine [13]. Parental vaccine hesitancy is complex and context-specific, varying across time, place, and vaccines, influenced by the factors such as convenience and confidence. Therefore, this study aimed to identify the factors related to parental hesitancy to vaccinate their children against COVID-19, using structured questionnaires additionally, and it sought to explore the reasons for parental COVID-19 vaccine hesitancy among factory workers through in-depth interviews. This study could provide a deeper understanding of the factors influencing parental

decision-making regarding the COVID-19 vaccine, supporting to develop the potential strategies for vaccination plans in the occupational setting.

## Materials and methods

### Study design, period, and setting

A study of cross-sectional sequential mixed-methods design was conducted, where a quantitative method was followed by a qualitative method, among parents of children under 16 who have not been vaccinated at Tri Star tyre factory, Ywama Industrial Zone, Insein township, Yangon Region, Myanmar, from August 2022 to February 2023.

### Phase 1: Quantitative phase

In phase 1, a quantitative survey was conducted through face-to-face structured questionnaires among factory workers to identify the factors related to parental hesitancy to vaccinate their children against COVID-19.

**Sample size and sampling method.** The sample size was determined by using a formula for the estimation of a single proportion [31], with 27% reporting parental hesitancy toward children under 18 years of age COVID-19 vaccination [32], 7% allowable error, and a 95% confidence interval. Then, the minimal required sample size was 155, which was optimized to 170 after adjusting the 10% non-response rate. The sampling frame was constructed after requesting the administrative office of the Tri Star tyre factory. All parents of children under 16-year-old working at Tri Star tyre factory were recruited for this study. The parents who were on leave at the time of data collection were excluded.

**Data collection tools and technique.** A pretested structured questionnaire was applied for the quantitative data collection involving face-to-face interviews (S1 File). It comprised four sections: 1) the socioeconomic characteristics of the parents, 2) the status of previous COVID-19 infection and vaccination, 3) attitude towards COVID-19 vaccination, and 4) the COVID-19 vaccine hesitancy. In the first section, there were 6 items related socioeconomic characteristics of the parents such as sex, age, level of education, occupational status, monthly family income, and total number of household members. The second section was the status of previous COVID-19 infection and vaccination which contained 6 items: previous COVID-19 infection of respondents, previous COVID-19 infection of partners, previous COVID-19 infection of children, vaccination status of respondents, vaccination status of partners, and main source of information on COVID-19 vaccine.

The third section, the attitude towards COVID-19 vaccination, was adapted from the COVID-Vaccination Attitude Scale (C-VAS) [33], which contained 16 items with the specific response options of 5-point Likert scale: strongly agree, agree, neutral, disagree and strongly disagree. The scores for positive attitude items were considered as 5 for strongly agree, 4 for agree, 3 for neutral, 2 for disagree, and 1 for strongly disagree. For the negative attitude item, scores for each item were designated reversely. In the fourth section, the parental COVID-19 vaccine hesitancy was assessed by the adopted form of the Oxford COVID-19 vaccine hesitancy scale (OC19-VHS), which contained 7 items coding 1–5 as an item-specific response options [32,34].

The items of the questionnaires for this study were prepared in consultation with the experts from public health to cover all the crucial aspects of parental vaccine hesitancy to COVID-19 observed in national context during the pandemic. The value of the content validity index was 0.97 for C-VAS and 1.00 for the OC19-VHS, based on six experts' ratings of item relevance. Then, the questionnaires were translated into the Myanmar language (S2 File). The pretest was carried out among 10% of the total required sample size, 20 parents in a garment factory, Mingaladon township, Yangon region. The reliability of the scale regarding the attitude towards COVID-19 vaccination was expressed as 0.86 of Cronbach's alpha coefficient.

**Operational definitions.** The educational status referred to the highest level of education that a participant has successfully completed and it was categorized as primary school level, middle school level, high school level, graduate

(completed bachelor degree), and postgraduate (completed advanced degrees such as master or doctoral study). Infected with SARS-CoV-2 was defined as the participants or his/her partners have been confirmed by laboratory test as positive. COVID-19 vaccination was defined as the participants or his/her partners having been vaccinated at least one dose of COVID-19 vaccine.

The attitude towards COVID-19 vaccination of the parents was defined as the importance of vaccination in control of disease spread, the safety of the vaccine, and concern about the side effects. The responses of the parents about COVID-19 vaccination through the attitude questionnaires answered with a median score and above were considered having positive attitude and those with below median score were considered to have negative attitude.

The parental COVID-19 vaccine hesitancy embodied the unwillingness to receive vaccines to the children when vaccination services are available and accessible. It was categorized into three groups: hesitant group (≥ 60% of the total score, ≥ 21), unsure group (41% to 59% of the total score, 15–20), and non-hesitant group (≤ 40% of the total score, ≤ 14).

**Quantitative data analysis.** The collected data were imported into Microsoft Excel 365 after checking the missing data. Then, the data were transferred to IBM SPSS Statistics for Windows, Version 26.0 (Armonk, NY: IBM Corp) for analysis. Descriptive statistics were used to describe the socioeconomic characteristics of the parents, status of previous COVID-19 infection and vaccination, level of attitude towards COVID-19 vaccination, and COVID-19 vaccine hesitancy. Numerical data were described using means and standard deviation (±SD) when approximately normally distributed and using median and inter-quartile range (IQR) when the data were skewed. Categorical variables were described as frequency, percentage, and confidence intervals (95% CI) for percentage.

For the final analysis, dependent variable was categorized as "parental COVID-19 vaccine hesitancy-yes" (vaccine hesitant group) and "parental COVID-19 vaccine hesitancy-no" (combined unsure and non-hesitant groups). To find out the predictors of parental COVID-19 vaccine hesitancy to their children, bivariate analysis was performed and all significant variables in the bivariate logistic regression analysis were retained in multivariate logistic regression analysis to determine the independent predictors of parental COVID-19 vaccine hesitancy. To assess the goodness of fit of the model, the Hosmer and Lemeshow test was used where $p \leq 0.05$ represented that the model was a poor fit. Sex and those who were infected with the SARS-CoV-2 were included in the multivariate logistic regression model. A p value $\leq 0.05$ was set up as a statistically significant for all analysis.

## Phase 2: Qualitative phase

**Participant selection.** A purposive sampling method was used for the qualitative method. After the quantitative analysis, a total of 12 participants, six participants from each category of parents' hesitancy to vaccinate their children under the age of 16 against COVID-19 (hesitant and non-hesitant group) were selected. To minimize selection bias, participants were selected from diverse backgrounds, including age, sex, educational status, and current department, to capture a wider range of perspectives. Fig 1 shows the flowchart of participant selection and data collection process of this study.

**Qualitative data collection.** In the qualitative approach, individual in-depth-interviews (IDIs) guides were explicitly adapted from earlier research for the study (S3 File), and probing each question provided new data from the parents' responses. The individual IDIs guide was constructed to explore the reasons for hesitancy of parents on COVID-19 vaccination to their children (for the parents from the hesitant group, n = 6) and the reasons for the non-hesitancy of parents on COVID-19 vaccination to their children (for the parents from the non-hesitant group, n = 6).

The parents were requested for the informed consent to participate in the interview, permission to audio-record, and used the information recorded as direct quotes in this study. The interviewing process was secured by choosing a quiet meeting room. After receiving informed consents from all participants, individual IDIs were conducted in a quiet meeting room during work break time. The length of the individual IDIs was a range of 15–30 minutes, and all interviews were audio-recorded for transcription purposes.

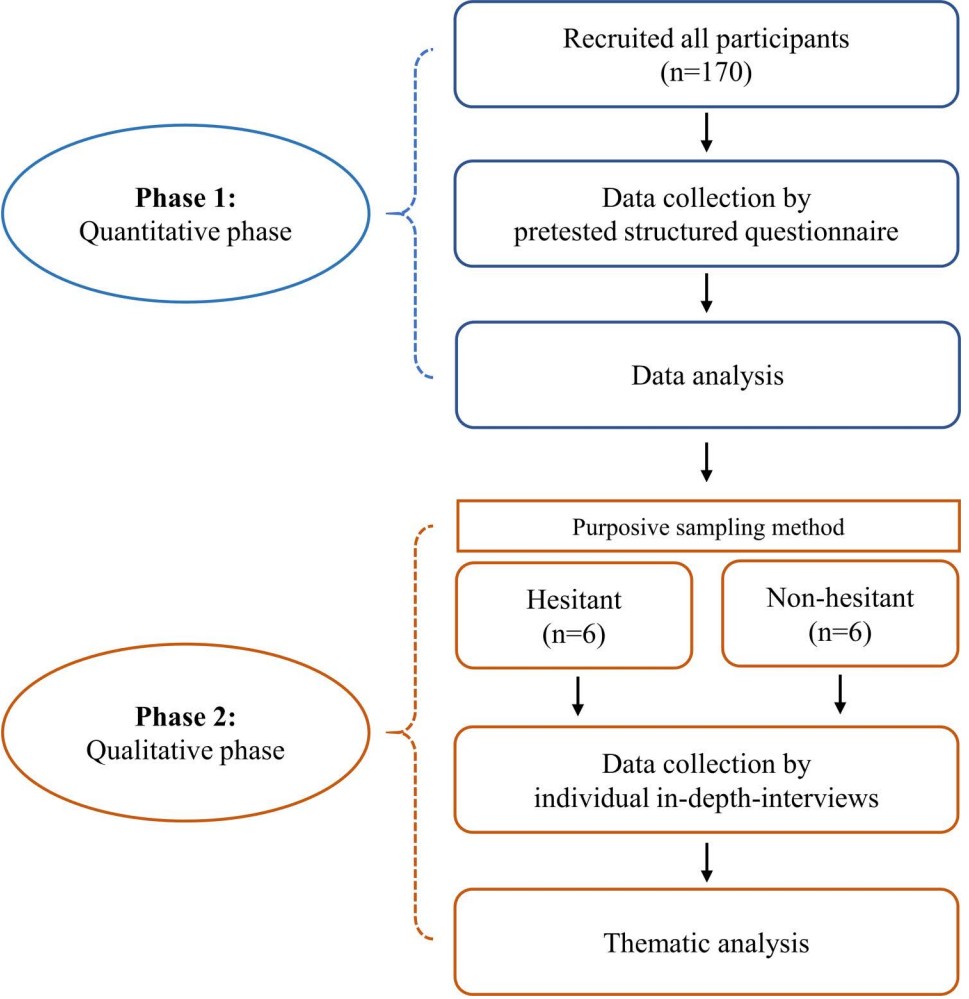

**Fig 1. Flowchart of participant selection and data collection process.**

**Qualitative data analysis.** After checking the integrity of the audio recordings, the data were transcribed into the Myanmar language. Verbatim records were translated into English with the help of a lecturer from the English Department of Defence Services Medical Academy. Then the check and recheck of audio recording and transcripts were undertaken by the supervisor and the familiarization of transcripts was completed by getting a thorough overview of all the collected data. The qualitative findings were analyzed by thematic analysis with the inductive approach. For the confused statement and conflict findings, consensus was taken by the supervisor before developing and refining a thematic framework.

The transcripts were then coded line-by-line using Microsoft Excel 365 and the text was color-highlighted as phrases or whole sentences to describe the context. A thematic schema was subsequently created with the generated themes, taking into consideration the research questions. The findings of the thematic analysis were presented with illustrative quotations of significant points, in which participants are identified by IDIs code number, age and sex.

**Ethical consideration.** Ethical clearance was requested from the Ethical Board and Postgraduate Board of Studies, Defence Services Medical Academy (Approval Number: DSMA/ERB/B-007/2022). The investigator met with the general manager of the Tri Star tyre factory to explain the aim of the study and the procedures and then took permission from the general manager. Participants were recruited according to inclusion and exclusion criteria and were invited to participate

in this study. Firstly, the participants were explained about the nature of the study and then written informed consent was taken from the participants before data collection. Data collection was conducted according to standard operating procedure of Ministry of Health (MOH) for COVID-19 precaution. All the information obtained from the respondents was confidential. After study, all participants detected as having parental hesitancy of COVID-19 vaccines to vaccinate their children were provided health education regarding the importance of COVID-19 vaccines to control the pandemic, benefits of getting vaccination, and the adverse effects of COVID-19 vaccines on human health.

## Results

### Phase 1: Quantitative results

In this study, a total of 170 parents from a factory were included in the analysis. The socio-economic characteristics of the participants were described in Table 1. Among the participants, 110 (64.7%) were female (mothers) and 60 (35.3%) were male (fathers). The mean (±SD) age of the participants was 33.89 (±6.34) years with a range from 20 to 62 years and 105 (61.8%) of the total participants were aged between 30 and 39 years. In the educational status, 109 (64.1%) participants graduated and only 2 (1.2%) passed the primary school level. Most of the participants, 118 (69.4%) were working in the production department, 37 (21.8%) were from the admin department, and 15 (8.8%) were from the management department. Of all participants, 167 (98.2%) were married and only 3 (1.8%) were widowed. The median (IQR) monthly family income of the participants was 317,500 (150,000) kyats [151.19 (71.43) USD] with a range of 150,000 [71.43 USD] to 900,000 kyats [428.57 USD] and 120 (70.6%) participants had 300,000 kyats [142.86 USD] and above monthly family income. The median (IQR) number of household member was 3 (1) with a minimum of 2 to a maximum of 7 and 103 (60.6%) participants had fewer than four family members.

Of all participants, 70 (41.2%) of the respondents were infected with SARS-CoV-2, 58 (34.1%) of the participants' partners were infected with SARS-CoV-2, and 21 (12.4%) of the participants' children who were infected with SARS-CoV-2 (Table 2). In the COVID-19 vaccination status, 167 (98.2%) of all participants and 163 (95.9%) of the participants' partners were vaccinated respectively. Fig 2 shows the main source of information on the COVID-19 vaccine among the participants: 119 (70.0%) of the respondents got information from social media, 94 (55.3%) from television, 56 (32.9%) from their friends, 51 (30.0%) from healthcare workers, 21 (12.4%) from newspapers, and 18 (10.6%) from radio.

Table 3 described the level of attitude towards COVID-19 vaccination and parental vaccine hesitancy among the participants. The median (interquartile range) attitude score was 61 (5) with the minimum 50 to maximum 79. The level of attitude towards COVID-19 vaccination was categorized as positive (median score and above) and negative (lower than the median score). In 170 respondents, 90 (52.9%, 95% CI: 45.1–60.6%) were positive attitudes and 80 (47.1%, 95% CI: 39.4–54.9%) were negative attitudes towards COVID-19 vaccination. For the COVID-19 vaccine hesitancy, 31 (18.2%, 95% CI: 12.7–24.9%) of the participants were hesitant to vaccinate their children against COVID-19, 44 (25.9%, 95% CI: 19.5–33.1%) were unsure for the vaccination, and 95 (55.9%, 95% CI: 48.1–63.5%) were in the non-hesitant group.

The factors associated with parental COVID-19 vaccine hesitancy are presented in Table 4. Males (30.0%) were 3.20 times more likely to have hesitancy than the females and sex was significantly associated with parental COVID-19 vaccine hesitancy (COR: 3.20, 95% CI: 1.44–7.12). Participants who had not been infected with the SARS-CoV-2 were 2.84 times likely to have hesitancy more than those who had been infected with the SARS-CoV-2 (COR: 2.84, 95% CI: 1.15–7.03). In the multivariate analysis, sex (AOR: 3.04, 95% CI: 1.35–6.84), and participants infected with the SARS-CoV-2 (AOR: 2.66, 95% CI: 1.06–6.70) remained at the significant associated factors of parental COVID-19 vaccine hesitancy.

### Phase 2: Qualitative findings

The qualitative interview explored the reasons for parental COVID-19 vaccine hesitancy to their children. The individual IDIs were performed to explore the reasons for hesitancy of parents on COVID-19 vaccination to their children from the

**Table 1. Socio-economic characteristics of the participants.**

| Variables | n (%) |
|---|---|
| Sex | |
| Male | 60 (35.3) |
| Female | 110 (64.7) |
| Age (year) | |
| < 30 | 39 (22.9) |
| 30-39 | 105 (61.8) |
| ≥ 40 | 26 (15.3) |
| Mean (± SD): 33.89 (± 6.34) years, Minimum 20 years, Maximum 62 years | |
| Educational status | |
| Primary school level | 2 (1.2) |
| Middle school level | 14 (8.2) |
| High school level | 42 (24.7) |
| Graduate | 109 (64.1) |
| Postgraduate | 3 (1.8) |
| Current department | |
| Administration | 37 (21.8) |
| Management | 15 (8.8) |
| Production | 118 (69.4) |
| Marital status | |
| Married | 167 (98.2) |
| Widowed | 3 (1.8) |
| Monthly family income * | |
| < 300,000 Kyats (< 142.86 USD) | 50 (29.4) |
| ≥ 300,000 Kyats (≥ 142.86 USD) | 120 (70.6) |
| Median (IQR): 317,500 (150,000, 250,000–400,000) kyat [151.19 (71.43, 119.05–190.48) USD], Minimum 150,000 kyats [71.43 USD], Maximum 900,000 kyats [428.57 USD] | |
| Number of household members | |
| < 4 | 103 (60.6) |
| ≥ 4 | 67 (39.4) |
| Median (IQR): 3 (1, 3–4), Minimum 2, Maximum 7 | |

*According to the reference exchange rate of the Central Bank of Myanmar, 1 USD to Myanmar Kyats is 2,100 during data collection period (https://forex.cbm.gov.mm).

hesitant group (n = 6) and the reasons for the non-hesitancy of parents on COVID-19 vaccination to their children from the non-hesitant group (n = 6).

**Reasons for hesitancy of parents on COVID-19 vaccination of children (n=6).** *Theme 1: Too young for vaccination*. The majority of the interviewees reported that young children might not fit to get vaccinated against COVID-19 because they were concerned about the harmful effects of the vaccine. Only a small number of interviewees indicated that they occurred fewer cases of children compared to the adult COVID-19 cases and there was no more required for vaccination to young children. One interviewee responded:

*"Because my child is too young and … even though he got a routine childhood vaccination, he felt sick. So, … I don't want to get a vaccine to my son because of the side effects that will make things worse."* (C19VH-5, 26 years old, male)

**Table 2. Status of previous COVID-19 infection and vaccination among participants.**

| Variables | n (%) |
| --- | --- |
| Infected with the SARS-CoV-2 | |
| Yes | 70 (41.2) |
| No | 100 (58.8) |
| Partner who infected with the SARS-CoV-2 (n = 167) | |
| Yes | 58 (34.1) |
| No | 109 (64.1) |
| Child who infected with SARS-CoV-2 | |
| Yes | 21 (12.4) |
| No | 149 (87.6) |
| COVID-19 vaccination | |
| Yes | 167 (98.2) |
| No | 3 (1.8) |
| Partner who got COVID-19 vaccination (n = 167) | |
| Yes | 163 (95.9) |
| No | 4 (2.4) |

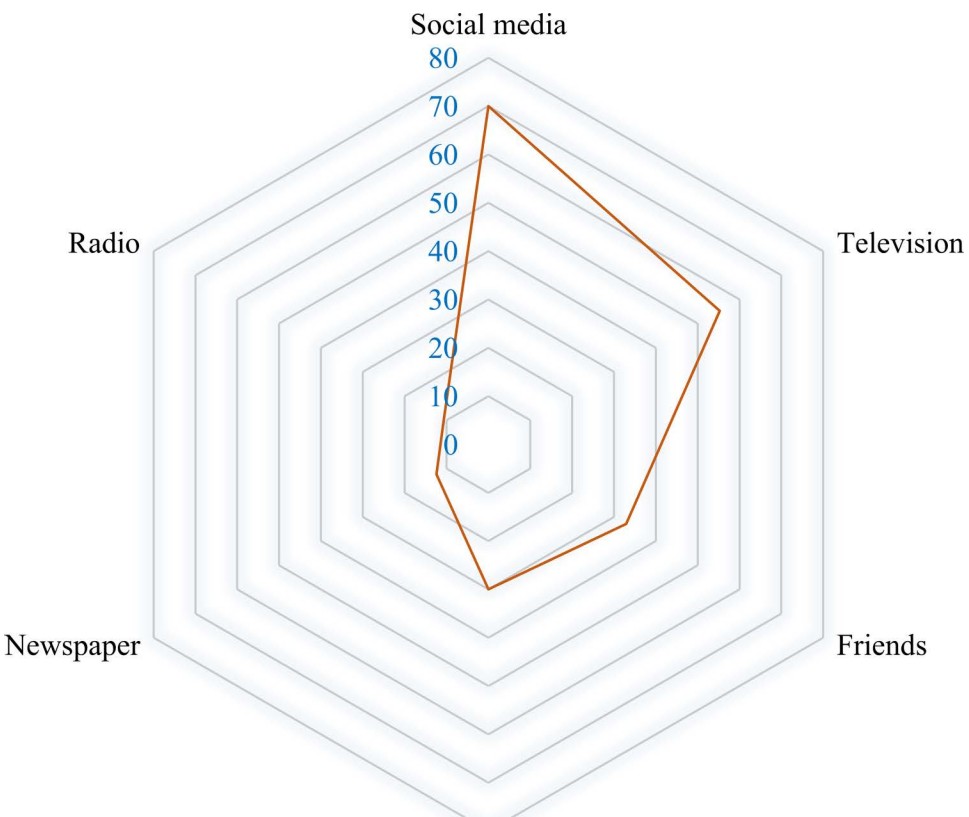

**Fig 2. Main sources of information on COVID-19 vaccine among participants.**

**Table 3. Attitude towards COVID-19 vaccination and Parental COVID-19 vaccine hesitancy among participants.**

| Variables | n (%) | 95% CI of percent |
|---|---|---|
| Level of attitude towards COVID-19 vaccination | | |
| Negative | 80 (47.1) | 39.4-54.9 |
| Positive | 90 (52.9) | 45.1-60.6 |
| Median (IQR): 61 (5, 59–64), Minimum 50, Maximum 79 | | |
| Parental COVID-19 vaccine hesitancy | | |
| Non-hesitant | 95 (55.9) | 48.1-63.5 |
| Unsure | 44 (25.9) | 19.5-33.1 |
| Hesitant | 31 (18.2) | 12.7-24.9 |
| Mean ± SD (14.25 ± 6.50), Minimum 7, Maximum 29 | | |

**Theme 2: Vaccine safety concern**. Almost two-thirds of the participants said that they did not believe and doubt about the safety of the COVID-19 vaccine due to rapidly developing and not enough research time. The most frequent reason for hesitant parents were unsure how safe for their children. Moreover, they worried that the vaccine made worse the health status of their children after getting vaccinated. However, a few parents were not concerned about the side effects and safety of the vaccine. One of the interviewees reported:

*"… I worry about the harmful effects of vaccination to my children. Even for adults, it is not easy to counter the side effects. And I am not sure how long the vaccine protects us."* (C19VH-5, 26 years old, male)

**Theme 3: Uncertainty of vaccine effectiveness**. The majority replied that there was no certainty regarding the effectiveness of the COVID-19 vaccine. A few parents responded that although they could not believe firmly in the COVID-19 vaccine if it would be mandatory to get vaccinated, they allowed their children to get vaccinated. As one interviewee said:

*"Of course, if my child contacts an infected person, he will be infected. I think … even thought if he got vaccinated, he could be infected." (C19VH-5, 26 years old, male)*

**Theme 4: Un-trust the origin of the vaccine**. The majority of respondents stated that the brand or origin of the COVID-19 vaccine could influence the decision to vaccinate their children. Additionally, consideration of safety for the children depended on the high-quality companies or country origins. However, a few parents replied that they were not aware of this issue for the COVID-19 vaccination. Interviewing about this issue, an interviewee stated:

*"It depends on the origin (company or country) of the COVID-19 vaccine. Actually…, I want to know the company and country where the vaccine comes from. It would be better if my child gets the high-quality vaccine. The government provided only one type of vaccine and we had no choice."* (C19VH-2, 30 years old, male)

**Reasons for non-hesitancy of parents on COVID-19 vaccination of children (n = 6).** **Theme 5: Necessity for their children.** The majority of the parents replied that they wanted to get the vaccine because COVID-19 vaccination would keep their children safe, protected, less contagious, and keep their children healthy. As one interviewee put it:

*"The vaccine is important for our children… because I know that prevention is more effective than treatment. Therefore, getting a COVID-19 vaccine is a good idea for everyone. Currently…, children go to school for their education and stay together in the classroom the whole day. No matter what their parents say, children might not wear masks all the time. I*

**Table 4. Associated factors of parental COVID-19 vaccine hesitancy among participants.**

| Variables | Parental COVID-19 vaccine hesitancy | | COR (95% CI) | p value | AOR (95% CI) | p value |
|---|---|---|---|---|---|---|
| | No | Yes | | | | |
| | n (%) | n (%) | | | | |
| **Sex** | | | | | | |
| Female | 97 (88.2) | 13 (11.8) | 1.00 | | 1.00 | |
| Male | 42 (70.0) | 18 (30.0) | 3.20 (1.44-7.12) | 0.004 | 3.04 (1.35-6.84) | 0.007 |
| **Age** | | | | | | |
| ≥ 30 | 111 (84.7) | 20 (15.3) | 1.00 | | | |
| < 30 | 28 (71.8) | 11 (28.2) | 2.18 (0.94-5.07) | 0.070 | | |
| **Education** | | | | | | |
| High school passed and below | 50 (86.2) | 8 (13.8) | 1.00 | | | |
| Graduate and above | 89 (79.5) | 23 (20.5) | 1.62 (0.67-3.88) | 0.283 | | |
| **Monthly family income** | | | | | | |
| < 300,000 Kyats (< 142.86 USD) | 99 (82.5) | 21 (17.5) | 1.00 | | | |
| ≥ 300,000 Kyats (≥ 142.86 USD) | 40 (80.0) | 10 (20.0) | 1.18 (0.51-2.72) | 0.701 | | |
| **Number of household members** | | | | | | |
| ≥ 4 | 58 (86.6) | 9 (13.4) | 1.00 | | | |
| < 4 | 81 (78.6) | 22 (21.4) | 1.75 (0.75-4.08) | 0.194 | | |
| **Infected with the SARS-CoV-2** | | | | | | |
| Yes | 63 (90.0) | 7 (10.0) | 1.00 | | 1.00 | |
| No | 76 (76.0) | 24 (24.0) | 2.84 (1.15-7.03) | 0.024 | 2.66 (1.06-6.70) | 0.037 |
| **Partner who infected with the SARS-CoV-2** | | | | | | |
| Yes | 51 (87.9) | 7 (12.1) | 1.00 | | | |
| No | 85 (78.0) | 24 (22.0) | 2.06 (0.83-5.11) | 0.121 | | |
| **Children infected with SARS-CoV-2** | | | | | | |
| Yes | 20 (95.2) | 1 (4.8) | 1.00 | | | |
| No | 119 (79.9) | 30 (20.1) | 5.04 (0.65-39.08) | 0.122 | | |
| **COVID-19 vaccination** | | | | | | |
| Yes | 138 (82.6) | 29 (17.4) | 1.00 | | | |
| No | 1 (33.3) | 2 (66.7) | 9.52 (0.84-108.50) | 0.070 | | |
| **Partner who got COVID-19 vaccination** | | | | | | |
| Yes | 133 (81.6) | 30 (18.4) | 1.00 | | | |
| No | 3 (75.0) | 1 (25.0) | 1.48 (0.15-14.70) | 0.739 | | |
| **Level of attitudes** | | | | | | |
| Positive | 78 (86.7) | 12 (13.3) | 1.00 | | | |
| Negative | 61 (76.3) | 19 (23.8) | 2.03 (0.91-4.50) | 0.083 | | |

COR: crude odds ratio, AOR: Adjusted odds ratio, Statistically significant level: p value ≤ 0.05

*think that if they are vaccinated, they may have a more powerful protection against COVID-19 than those who are not."* (C19VH-10, 28 years old, female)

***Theme 6: Improvement of immunity against COVID-19.*** Most parents replied that their children should get the vaccine due to an improved immune system. A small number of interviewees suggested that COVID-19 vaccination was not related to the improvement of immunity in children. One participant reported:

*"I want to get my child vaccinated against COVID-19 because it provides better protection against COVID-19 than not getting vaccinated."* (C19VH-11, 34 years old, female)

**Theme 7: Decreasing the COVID-19 transmission and death rate.** The majority of the parents replied that COVID-19 vaccination can reduce the new case detection and death. This was a reason that they want their children to get vaccinated, and one of the interviewees said:

*"The new COVID-19 cases and deaths can be reduced by vaccinations in the community, and then also reduce the spread to others. That is… why I want my child to get vaccinated."* (C19VH-7, 33 years old, female)

**Theme 8: Cost of vaccine**. Majority of the parents reported that the cost of vaccination against COVID-19 did not rely on their willingness to vaccinate their children. They would pay for the vaccine if it was truthy necessary. One informant described:

*"I want my daughter to get vaccinated actually… but it does not depend on the cost of the vaccine. I will pay for it if the vaccine is truly protecting my daughter from COVID-19."* (C19VH-12, 42 years old, female)

**Theme 9: Safety and effectiveness of vaccine.** The majority of the interviewees replied that they wanted to get vaccinated for their children because they believed in the safety and efficacy of the vaccines. Only a small number of parents indicated that the main reason for the child was vaccinated against COVID-19 was to protect him from infection, not because of the vaccine's efficacy. As one interviewee commented:

*"I believe that we don't need to worry about the ineffectiveness of vaccine if we follow the recommended guideline and complete doses of COVID-19 vaccination to our children."* (C19VH-9, 37 years old, female)

## Discussion

With the expansion of the COVID-19 vaccine program to children, it becomes more vital to understand the factors linked to parental vaccine hesitancy and COVID-19 vaccines. This study examined the parental hesitancy to vaccinate their children under the age of 16 against COVID-19 among factory workers. Among the total, (41.2%) of the respondents were infected with COVID-19, and (34.1%) of the respondents' partners were infected with SARS-CoV-2 since the beginning of the pandemic. In contrast, a study in Saudi Arabia showed that (38.0%) of adults in the households were infected with COVID-19 [23]. In another similar study conducted in Bangladesh, (28.6%) of parents revealed that they or their family members had COVID-19 positive tests [35].

The current study found that (12.4%) of respondents' children were infected with SARS-CoV-2, and a similar study performed in Saudi Arabia showed that (13.4%) of parents had revealed a history of laboratory-confirmed COVID-19 in their children [32]. Moreover, it was lower than the finding of another previous Saudi Arabia study (21.5%) [23]. This inconsistency of previous COVID-19 infection might be due to differences in healthcare infrastructure, demographic characteristics, variability of COVID-19 epidemic burden in study areas, and effectiveness of mitigation measures of government organizations. Another explanation might be decreasing trend of COVID-19 transmission and the effect of COVID-19 vaccination programs.

In this study, the majority of participants (98.2%) and their partners (95.9%) were vaccinated against COVID-19. The previous studies stated that (81.1%) of the participants had already received at least one dose of the COVID-19 vaccine [32] and (95.8%) of the respondents were vaccinated against COVID-19 [23]. It showed that hesitancy to COVID-19 vaccination was decreasing in adults and vaccination rates tend to be increasing. Factors such as the socioeconomic status

of the countries, level of healthcare infrastructure, ability to control the spread of infection, and self-protection and risk perception during the pandemic could influence the vaccination rate.

In this study, the most trusted source of information for the respondents regarding COVID-19 vaccination was social media (70.0%), followed by television (55.3%). This study was consistent with those conducted in Saudi Arabia, where parents mainly used social media and the official websites of the MOH to learn more about the COVID-19 vaccine [23]. In a Japanese study, the most reliable sources of information about the COVID-19 vaccine were government and public organizations (26.7%), followed by the private news media (22.3%) [36]. These findings supported for the strengthening of the utilization of social media and official platforms of government for risk communication regarding COVID-19 vaccination program.

The attitude towards COVID-19 vaccination was critical for the government and policymakers to address the obstacles in implementing the COVID-19 vaccination program. In this study, (47.1%) of the parents were at the level of negative attitude toward COVID-19 vaccination and it was lower than the finding of a study conducted in Ethiopia stated that (55.3%) of parents had negative attitudes towards COVID-19 vaccine [37]. Conversely, it was higher than the findings of the Saudi Arabia study (27.6%) [38] and Bangladesh (22.0%) [39]. A Thailand study provided that the participants had high levels of positive attitude towards the COVID-19 vaccine and attitude towards the vaccination program [40]. These discrepancies regarding the attitude towards COVID-19 vaccination might be due to the differences in study areas, measurement tools and their cut-off points, awareness of disease burden, and previous COVID-19 infection. To increase vaccination coverage, people should develop a more positive attitude toward the importance of vaccines that can reduce the disease severity and control the disease spread in the community.

For the COVID-19 vaccine hesitancy, (18.2%) of the participants were hesitant to vaccinate their children against COVID-19 in this study. This finding was higher than a study done in USA stated (15.1%) of participants had COVID-19 vaccine hesitancy to their children [24]. However, it was lower than the proportions of parental vaccine hesitancy against COVID-19 to get their children reported in most of the previous studies: 20.3% in Romania [27], 28.7% in Ireland [41], (26.7%) and (61.9%) in Saudi Arabia [23,32], 35.3% in Japan [36], (28.9–46.2%) in the USA [25,42–44], (33.1%) in Singapore [28], and (56.5%) in Ohio [22]. The acceptance of vaccinations is influenced by public confidence in institutions like the government and the health care system. The inconsistency of parental COVID-19 vaccine hesitancy might be due to the differences in education level, previous COVID-19 infection, experience about vaccination among parents, assessment tools, concern about the value or safety of vaccination, trust of information about the vaccine, and fear of side-effect.

The hesitant parents replied that being too young for vaccination, concern about the safety of the COVID-19 vaccine, uncertainty of vaccine effectiveness, and untrusting the origin of the vaccine were the main reasons for the hesitancy to vaccinate their children against COVID-19. In a mixed-methods study, the parents also reported that in the COVID-19 vaccination for children, age might be more harmful than the COVID-19 infection itself. The parents worried about immediate, and unexpected adverse effects [44]. The parents were also concerned about the COVID-19 vaccines, which might not be as good in terms of quality or efficacy due to the short time trial period before they were released for use [45]. Additionally, the brand or origin of the COVID-19 vaccine influenced the decision to vaccinate their children. The parents would more willingness to vaccinate their children with the COVID-19 vaccine that has been tested and proven safe.

In this study, there was a statistically significant association between sex and parental COVID-19 vaccine hesitancy. Contrary to expectation, fathers were more likely to have hesitancy than mothers in this study and it reflected the finding of a previous study done in Singapore [28]. In Myanmar, most of fathers hold primary authority, and were considered as the head of the household, making the main decision for the family. The earlier studies done in Japan [36], Saudi Arabia [23], Ireland [41] and USA [42] stated that mothers were more likely to be hesitant about COVID-19 vaccination than fathers and this was significant. However, this finding did not support the previous researches reported that sex was not an associated factor of parental COVID-19 vaccine hesitancy [43,46].

There was a significant association between previous SARS-CoV-2 infection and parental COVID-19 vaccine hesitancy in the current study. The parents with no history of previous SARS-CoV-2 infection were likelihood of vaccine hesitancy to their children and this finding supported to the previous study [47]. It was possible that uninfected parents had poor concern about the severity of COVID-19 disease and the perception that vaccination could reduce the disease transmission, severity, and mortality. Nevertheless, a Saudi Arabia study showed that adults in the household with a history of previous COVID-19 infection were more likely to be unwilling to vaccinate their children [23]. Another study also found that participants who had previously been infected with COVID-19 had higher parental COVID-19 vaccine hesitancy than those who had not been infected [32].

The current study did not support the findings of similar studies, stated that the agreement of parents to their children for the COVID-19 vaccination differed significantly in terms of parental age groups [28,41,43,47,48]. Moreover, no differences in the parental COVID-19 vaccine hesitancy to their children regarding education status occurred in the current study. However, this finding was contrary to previous studies which had suggested that parents with higher education levels were less likely to vaccinate their children than parents with education levels below high school [23,28,49]. A Saudi Arabia study also described that a determinant of the hesitancy to childhood COVID-19 vaccination was the parents with an education level of a university degree or above compared to those having a lower education level [32]. In this study, family income was not associated with parental COVID-19 vaccine hesitancy. This finding differed from some published studies, which mentioned that household income was a significant associated factor for parental COVID-19 vaccine hesitancy to their children [25,28,44].

There was no association between previous SARS-CoV-2 infection of participants' children and parental COVID-19 vaccine hesitancy. This result matched those observed in the previous study conducted in Saudi Arabia, which showed that previous COVID-19 infection of children was not associated with parental COVID-19 vaccine hesitancy [23]. It was difficult to explain this result, but it might be related to the pattern and severity of the COVID-19 infection, in which infected children had mild symptoms compared with adults. One unanticipated finding was no significant relationship between participants' COVID-19 vaccination and parental COVID-19 vaccine hesitancy in this study. However, the findings of the current study do not support the previous research, which found that parents who want to vaccinate themselves also plan to vaccinate their children [36]. Another study in Saudi Arabia showed that vaccinated parents would be more willing to vaccinate their children [32].

Regarding the attitude towards COVID-19 vaccine among the parents, this study did not find a significant difference between parental COVID-19 vaccine hesitancy and attitudes towards COVID-19 vaccination. In contrast to earlier findings in Saudi Arabia, parents allowed COVID-19 vaccine to their children had higher attitude scores than those whose children did not receive the vaccine. Most children did not receive the COVID-19 vaccine because of parental concerns about the side effects of the vaccine [50]. In a previous study done in Ohio revealed that COVID-19 vaccination attitude was associated with intention to vaccinate to the children. Most of parents who had high COVID-19 vaccination attitude scale did not intend to vaccinate to their children [22]. It was possible that these results are due to the differences in educational status, level of health literacy and previous COVID-19 infection and vaccination of the participants.

This study was conducted as a mixed-methods approach to understand the holistic picture of parental COVID-19 vaccine hesitancy to their children, which added depth and breadth to the study. A quality assurance strategy was implemented to detect and correct the errors or inconsistencies in both quantitative data and qualitative findings. Particularly in a pandemic context, the interviews were undertaken with the guidance of the standard operating procedure of the MOH to prevent the transmission of SAR-CoV-2. Even with its strengths, there were some limitations in this current study. First, due to the cross-sectional nature of the study, it could not establish a causal relationship. Second, this study was a single-center study, which impedes the generalizability of the findings and hence, further research would be needed to conduct with a large sample as a multi-centers study for the stronger association of independent predictors with parental COVID-19 vaccine hesitancy. Third, the qualitative part was only to explore the reasons for parental

COVID-19 vaccine hesitancy to their children. Hence, the quantitative and qualitative parts could not be triangulated in this study.

## Conclusion

In conclusion, this study identified that nearly one-fifth of the participants were hesitant the vaccination against COVID-19 to their children. Sex and previous SARS-CoV-2 infection of participants were significantly associated with parental COVID-19 vaccine hesitancy to their children. Too young for the vaccination, vaccine safety concern, uncertainty of vaccine effectiveness and untrust in the origin of the vaccine were the main reasons for hesitancy to get vaccinated to their children. In order to improve the National Deployment and Vaccination Plan, the real-time exchange of information about COVID-19 vaccine, advice and opinions should be done by using social media and television program. Health education programs about COVID-19 vaccination and effective vaccination campaigns, along with announcements of vaccine information regarding safety, efficacy and effectiveness should be provided to improve the proportion of children getting vaccinated against COVID-19.

## Supporting information

**S1 File. This is the S1 File English version of the questionnaire.**
(PDF)

**S2 File. This is the S2 File Myanmar version of the questionnaire.**
(PDF)

**S3 File. This is the S3 File Individual in-depth interview guide.**
(PDF)

**S4 File. This is the S4 File Minimal raw data.**
(XLSX)

## Acknowledgments

We thank to all participants for their voluntary participation in this study. We would like to express our heartfelt thanks to the general manager of Tri Star tyre factory for permitting to perform this study in factory and his kind support.

## Author contributions

**Conceptualization:** Kyaw Thiha Aung, Ye Minn Htun, Tun Tun Win.

**Data curation:** Kyaw Thiha Aung, Ye Minn Htun, May Soe Aung.

**Formal analysis:** Kyaw Thiha Aung, Ye Minn Htun.

**Investigation:** Kyaw Thiha Aung, Zin Lin Htet, Phyo Ko Ko.

**Methodology:** Kyaw Thiha Aung, Ye Minn Htun, Win Oo, May Soe Aung, Tun Tun Win.

**Supervision:** Ye Minn Htun, Tun Tun Win.

**Validation:** Kyaw Thiha Aung, Ye Minn Htun, Yan Naing Myint Soe.

**Visualization:** Kyaw Thiha Aung, Ye Minn Htun.

**Writing – original draft:** Kyaw Thiha Aung, Ye Minn Htun.

**Writing – review & editing:** Kyaw Thiha Aung, Ye Minn Htun, Zin Lin Htet, Yan Naing Myint Soe, Phyo Ko Ko, Win Oo, May Soe Aung, Tun Tun Win.

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
