## [Decision Letter · Decision Letter 0]

PONE-D-25-10068Parental hesitancy on COVID-19 vaccination of children under the age of 16: A cross-sectional mixed-methods study among factory workersPLOS ONE

Dear Dr. Htun,

Thank you for submitting your manuscript to PLOS ONE. After careful consideration, we feel that it has merit but does not fully meet PLOS ONE’s publication criteria as it currently stands. Therefore, we invite you to submit a revised version of the manuscript that addresses the points raised during the review process.

This manuscript is in need of a major revision. Thus, please address our reviewers' concerns and improve this manuscript. Thank you!

We look forward to receiving your revised manuscript.

Kind regards,

Harapan Harapan, MD, PhD

Academic Editor

PLOS ONE

3. We note that there is identifying data in Table 5. Prior to sharing human research participant data, authors should consult with an ethics committee to ensure data are shared in accordance with participant consent and all applicable local laws.

-Location data

Please remove or anonymize all personal information, ensure that the data shared are in accordance with participant consent, and re-upload a fully anonymized data set. Please note that spreadsheet columns with personal information must be removed and not hidden as all hidden columns will appear in the published file.

Reviewers' comments:

Reviewer's Responses to Questions

**Comments to the Author**

1. Is the manuscript technically sound, and do the data support the conclusions?

Reviewer #2: Partly

Reviewer #3: Yes

2. Has the statistical analysis been performed appropriately and rigorously? 

Reviewer #2: Yes

Reviewer #3: Yes

3. Have the authors made all data underlying the findings in their manuscript fully available?

Reviewer #2: Yes

Reviewer #3: Yes

4. Is the manuscript presented in an intelligible fashion and written in standard English?

Reviewer #2: No

Reviewer #3: Yes

5. Review Comments to the Author

Reviewer #2: A review report of the manuscript entitled “Parental hesitancy on COVID-19 vaccination of children under the age of 16: A cross- sectional mixed-methods study among factory workers”

- (page 3, lines 43-44) The opening sentence would be better to rewrite this way: “Thanks to the development of COVID-19 vaccines, now they can be safely and effectively used to guard COVID-19 patients against severe illness, hospitalization, and even mortality.”

- (page 3, line 53) "A total of 170 factory workers having children under the age of 16 in this study" is incomplete. You could say, "A total of 170 factory workers with children under the age of 16 participated in this study."

- (page 5, lines 86-88) This sentence “COVID-19 is continuing to spread around the world with more than 758 million confirmed cases and more than 6.8 million deaths...” also shares the same idea with a similar study by Duta et al. entitled “Essential oil for COVID-19 management: A systematic review of randomized controlled trials” Kindly include this reference to make the sentence stronger.

- (page 5, line 105) As authors have defined the acronym of WHO earlier in this manuscript (line 83), there’s no need to give its full definition anymore on the next mention. So, using only WHO here is already enough.

- (page 6, lines 112-121) This paragraph “Vaccine hesitancy, which expresses the unwillingness to get vaccines when vaccination services are available and accessible, is one of the greatest threats to global health…” will be more significant if it’s also supported by another relevant study. For example, from Hassan et al. entitled “Global acceptance and hesitancy of COVID-19 vaccination: A narrative review”

- (page 8, lines 159-160) The sentence "All parents having children under 16-year-old were working at…" is grammatically incorrect. I would suggest writing it this way "All parents of children under 16 years old working at Tri Star Tyre Factory were recruited for this study."

- (page 8, lines 166-168) The authors stated that “In the first section, there were 6 items related socioeconomic characteristics of the parents such as sex, age, level of education, occupational status, monthly family income, and family members.” My concern is what kind of data has been taken from “family members”. Was that the number of them in a family? Please clarify.

- Authors are suggested to proofread the manuscript after addressing all comments to avoid any typological, grammatical, and lingual mistakes and errors. For example, “questionaries” on page 9 line 182; full stop missing on page 10 line 216; “their children received the had higher attitude scores” on page 27 line 526.

- (page 11, line 231) Please give the meaning of IDIs abbreviation on the first mention.

- (page 13, lines 279-280) “In the educational status, (64.1%) participants graduated…” Do the authors mean “bachelor graduate” here?

- (page 13, lines 284-285) “…monthly family income of the participants was 317,500 (150,000) kyats with a range of 150,000 to 900,000 kyats…” It would be clearer if authors gave the US dollar estimation of the salary received by the factory workers here. So please add the amount of the money they receive as salary in both kyats and USD.

- (page 23, lines 430-431) “It showed that acceptance of COVID-19 vaccine hesitancy was increasing in adults and vaccination rates tend to be increasing.” This sentence contradicts the previous sentences as the authors used the “hesitancy” word in it. It should be revised.

- (page 23, lines 435-436) “In this study, the most trusted source of information for the respondents regarding COVID-19 vaccination was social media (70.0%), followed by television (55.3%).” This is totally new information, as authors did not mention/report it previously in the methods and results sections.

Reviewer #3: Thank you for the opportunity to review your manuscript entitled “Parental Hesitancy on COVID-19 Vaccination of Children Under the Age of 16: A Cross-Sectional Mixed-Methods Study Among Factory Workers.” I very much enjoyed reading your manuscript.

Summary:

This study investigates parental hesitancy toward COVID-19 vaccination for children under 16 among factory workers in Myanmar, utilizing a cross-sectional mixed-methods design. Quantitative data were gathered through structured interviews with 170 factory workers and analyzed using binary logistic regression to determine factors associated with vaccine hesitancy. The qualitative component involved in-depth interviews with 12 participants to explore reasons for hesitancy and non-hesitancy. The results indicate that 18.2% of parents were hesitant, 25.9% were unsure, and 55.9% were non-hesitant. Key factors associated with hesitancy included male gender (AOR: 3.04, 95% CI: 1.35–6.84) and lack of prior SARS-CoV-2 infection (AOR: 2.66, 95% CI: 1.06–6.70). Common concerns included child age, vaccine safety, uncertainty about effectiveness, and distrust in vaccine origin. The study suggests that health education programs and media campaigns should be strengthened to address misinformation and improve vaccination rates. While the study provides valuable insights into vaccine hesitancy within a specific workforce, further multi-center studies with a larger sample size and comparative analysis across different occupational groups could enhance its generalizability.

Major Comments:

1. Literature Review:

- You may expand the discussion on global vaccine hesitancy trends, particularly in occupational settings, to contextualize the study findings.

- You may provide more comparative insights into vaccine hesitancy among factory workers versus the general population in Myanmar or similar settings.

- You may include recent studies on COVID-19 vaccine acceptance among parents to strengthen the theoretical background.

2. Theoretical Contribution:

- You may clearly define how this study adds to the existing body of knowledge on vaccine hesitancy in occupational settings.

- You may highlight the novelty of the findings. Does the study reveal new factors influencing hesitancy, or does it align with prior research?

3. Methods:

- You may clarify how participants were selected for in-depth interviews and whether selection bias may have influenced the qualitative findings.

4. Results:

- You may include a comparative table of hesitant versus non-hesitant groups to highlight key differences in demographics and attitudes.

5. Figures:

- Figure 1: An in-text citation to is missing. You may mention where it is cited in the text.

- You may include a flow diagram of participant selection to improve transparency.

- You may ensure that figure numbering matches in-text citations.

Minor Comments:

1. Abbreviations: You may define abbreviations at their first mention (e.g., IDIs at line 231).

2. Title: You may consider adding “A Pilot Study” at the end of the title, as the study sample is relatively small.

3. Tables:

- You may add a footnote explaining abbreviations and statistical significance levels used in the tables.

- Table 4: You may ensure consistent formatting of the table and fill every cell, if applicable.

6. PLOS authors have the option to publish the peer review history of their article (what does this mean? ). If published, this will include your full peer review and any attached files.

**Do you want your identity to be public for this peer review?** For information about this choice, including consent withdrawal, please see our Privacy Policy .

Reviewer #2: No

Reviewer #3: **Yes: ** Ibrahim Saleh

---

## [Author Response · Author response to Decision Letter 1]

14 Apr 2025

Responses to editor and reviewers

PONE-D-25-10068R1

Parental hesitancy on COVID-19 vaccination of children under the age of 16: A cross-sectional mixed-methods study among factory workers

PLOS ONE

Revised manuscript on April 14, 2025

Dear Prof. Harapan Harapan,

I would like to thank you for the opportunity to revise and resubmit our manuscript titled “Parental hesitancy on COVID-19 vaccination of children under the age of 16: A cross-sectional mixed-methods study among factory workers”. We are grateful to you and the reviewers for their helpful comments on our manuscript. It is our belief that the manuscript is substantially improved after making the suggested edits.

Followings this letter are the editor and reviewers’ comments with our responses in point by point, including how and where the text was modified. Changes made in the revised manuscript are marked using track changes. The revision has been developed in consultation with all authors, and each author has given approval to the final form of this revision.

Thank you again for consideration of our revised manuscript.

Sincerely,

Ye Minn Htun

Health and Disease Control Unit

Nay Pyi Taw 15011, Myanmar

Journal Requirements

Comment 1: Please ensure that your manuscript meets PLOS ONE's style requirements, including those for file naming. The PLOS ONE style templates can be found at

Response: Thanks for this comment and we made sure that our manuscript meets PLOS ONE’s style requirements.

Comment 2: Please provide additional details regarding participant consent. In the ethics statement in the Methods and online submission information, please ensure that you have specified what type you obtained (for instance, written or verbal, and if verbal, how it was documented and witnessed). If your study included minors, state whether you obtained consent from parents or guardians. If the need for consent was waived by the ethics committee, please include this information.

Response: Thank you for this comment and the manuscripts conformed the STROBE guideline for the observational studies. We revised in manuscript and “Ethics Statement” field of the submission form as the following:

“Firstly, the participants were explained about the nature of the study and then written informed consent was taken from the participants before data collection.” (Materials and methods, Page No. 13, Line No. 287-288)

Comment 3: We note that there is identifying data in Table 5. Prior to sharing human research participant data, authors should consult with an ethics committee to ensure data are shared in accordance with participant consent and all applicable local laws.

-Location data

Please remove or anonymize all personal information, ensure that the data shared are in accordance with participant consent, and re-upload a fully anonymized data set. Please note that spreadsheet columns with personal information must be removed and not hidden as all hidden columns will appear in the published file.

Response: Thanks for your suggestion. We revised and re-upload the Excel data set file (S4 file) after removing all personal information. Moreover, concerning for the direct or indirect identifiers of data, we removed Table 5 and revised as the following:

“The individual IDIs were performed to explore the reasons for hesitancy of parents on COVID-19 vaccination to their children from the hesitant group (n=6) and the reasons for the non-hesitancy of parents on COVID-19 vaccination to their children from the non-hesitant group (n=6).” (Results, Page No. 19, Line No. 348-351)

Review Comments to the Author

Reviewer #2:

A review report of the manuscript entitled “Parental hesitancy on COVID-19 vaccination of children under the age of 16: A cross- sectional mixed-methods study among factory workers”

Comment 1: (page 3, lines 43-44) The opening sentence would be better to rewrite this way: “Thanks to the development of COVID-19 vaccines, now they can be safely and effectively used to guard COVID-19 patients against severe illness, hospitalization, and even mortality.”

Response: Thanks for your suggestion and we rewrite the sentence. (Abstract, Page No. 3, Line No. 43-45)

Comment 2: (page 3, line 53) "A total of 170 factory workers having children under the age of 16 in this study" is incomplete. You could say, "A total of 170 factory workers with children under the age of 16 participated in this study."

Response: We would like to thank for this comment. And we revised as follow:

“A total of 170 factory workers with children under the age of 16 participated in this study.” (Abstract, Page No. 3, Line No. 54)

Comment 3: (page 5, lines 86-88) This sentence “COVID-19 is continuing to spread around the world with more than 758 million confirmed cases and more than 6.8 million deaths...” also shares the same idea with a similar study by Duta et al. entitled “Essential oil for COVID-19 management: A systematic review of randomized controlled trials” Kindly include this reference to make the sentence stronger.

Response: Thanks for your suggestion. We agreed with it and revised as following:

“COVID-19 is continuing to spread around the world with more than 758 million confirmed cases and more than 6.8 million deaths, at the end of February 2023 [2, 3].” (Introduction, Page No. 5, Line No. 87-89)

Comment 4: (page 5, line 105) As authors have defined the acronym of WHO earlier in this manuscript (line 83), there’s no need to give its full definition anymore on the next mention. So, using only WHO here is already enough.

Response: Thanks for this positive comment. We edited it as follow:

“WHO recommended that the countries should continue to accomplish …” (Introduction, Page No. 5, Line No. 106-107)

Comment 5: (page 6, lines 112-121) This paragraph “Vaccine hesitancy, which expresses the unwillingness to get vaccines when vaccination services are available and accessible, is one of the greatest threats to global health…” will be more significant if it’s also supported by another relevant study. For example, from Hassan et al. entitled “Global acceptance and hesitancy of COVID-19 vaccination: A narrative review”

Response: Thank you for this suggestion and we added this relevant study for the more significant information.

“Vaccine hesitancy, which expresses the unwillingness to get vaccines when vaccination services are available and accessible, is … suffering and deaths [14, 15].” (Introduction, Page No. 6, Line No. 112-121)

Comment 6: (page 8, lines 159-160) The sentence "All parents having children under 16-year-old were working at…" is grammatically incorrect. I would suggest writing it this way "All parents of children under 16 years old working at Tri Star Tyre Factory were recruited for this study."

Response: We would like to thank for it and revised as follow:

“All parents of children under 16-year-old working at Tri Star tyre factory were recruited for this study.” (Materials and Methods, Page No. 8, Line No. 172-173)

Comment 7: (page 8, lines 166-168) The authors stated that “In the first section, there were 6 items related socioeconomic characteristics of the parents such as sex, age, level of education, occupational status, monthly family income, and family members.” My concern is what kind of data has been taken from “family members”. Was that the number of them in a family? Please clarify.

Response: Thanks for the comment to clarify. We mean that this variable is the total number of individuals who are residing in the same housing unit. Therefore, to get more clarify, we edited as follow:

“In the first section, there were 6 items related socioeconomic characteristics of the parents such as sex, age, level of education, occupational status, monthly family income, and total number of household members.” (Materials and Methods, Page No. 8, Line No. 179-182)

AND

Table 4. (Results, Page No. 17-18, Line No. 344)

Comment 8: Authors are suggested to proofread the manuscript after addressing all comments to avoid any typological, grammatical, and lingual mistakes and errors. For example, “questionaries” on page 9 line 182; full stop missing on page 10 line 216; “their children received the had higher attitude scores” on page 27 line 526.

Response: Thanks for pointing them out. We re-checked throughout the manuscript and revised according to your comments as followings:

“The items of the questionnaires for this study were prepared …” (Materials and Methods, Page No. 9, Line No. 196)

“…, and “parental COVID-19 vaccine hesitancy-no” (combined unsure and non-hesitant groups).” (Materials and Methods, Page No. 11, Line No. 233-234)

“..., parents allowed COVID-19 vaccine to their children had higher attitude scores than those whose children did not receive the vaccine.” (Discussion, Page No. 27, Line No. 548-550)

Comment 9: (page 11, line 231) Please give the meaning of IDIs abbreviation on the first mention.

Response: Thanks for this comment and we revised as following:

“In the qualitative approach, individual in-depth-interviews (IDIs) guides were explicitly adapted …” (Materials and Methods, Page No. 11, Line No. 253)

Comment 10: (page 13, lines 279-280) “In the educational status, (64.1%) participants graduated…” Do the authors mean “bachelor graduate” here?

Response: Thank you for the comment. Graduate in the educational status means the participant who has completed bachelor degree. To more clarify, we added the operational definition of educational status as following:

“The educational status referred to the highest level of education that a participant has successfully completed and it was categorized as primary school level, middle school level, high school level, graduate (completed bachelor degree), and postgraduate (completed advanced degrees such as master or doctoral study).” (Materials and Methods, Page No. 9-10, Line No. 205-208)

Comment 11: (page 13, lines 284-285) “…monthly family income of the participants was 317,500 (150,000) kyats with a range of 150,000 to 900,000 kyats…” It would be clearer if authors gave the US dollar estimation of the salary received by the factory workers here. So please add the amount of the money they receive as salary in both kyats and USD.

Response: Thanks for this suggestion. We agreed with this suggestion and described both kyats and USD as following:

“The median (IQR) monthly family income of the participants was 317,500 (150,000) kyats [151.19 (71.43) USD] with a range of 150,000 kyats [71.43 USD] to 900,000 kyats [428.57 USD] and 120 (70.6%) participants had 300,000 kyats [142.86 USD] and above monthly family income.” (Results, Page No. 14, Line No. 303-306)

And remarked in Table 1 as:

“According to the reference exchange rate of the Central Bank of Myanmar, 1 USD to Myanmar Kyats is 2,100 during data collection period (https://forex.cbm.gov.mm).”

Comment 12: (page 23, lines 430-431) “It showed that acceptance of COVID-19 vaccine hesitancy was increasing in adults and vaccination rates tend to be increasing.” This sentence contradicts the previous sentences as the authors used the “hesitancy” word in it. It should be revised.

Response: Thanks for your pointing out and we edited as following:

“It showed that hesitancy to COVID-19 vaccination was decreasing in adults and vaccination rates tend to be increasing.” (Discussion, Page No. 23, Line No. 453-454)

Comment 13: (page 23, lines 435-436) “In this study, the most trusted source of information for the respondents regarding COVID-19 vaccination was social media (70.0%), followed by television (55.3%).” This is totally new information, as authors did not mention/report it previously in the methods and results sections.

Response: Thanks for the comment and we have mentioned “the main sources of information” in results section as followings:

“… vaccination status of partners, and main source of information on COVID-19 vaccine.” (Materials and Methods, Page No. 9, Line No. 185-186)

“Fig 2 shows the main source of information on the COVID-19 vaccine among the participants: 119 (70.0%) of the respondents got information from social media, 94 (55.3%) from television, …” (Results, Page No. 15, Line No. 316-319)

Reviewer #3:

Thank you for the opportunity to review your manuscript entitled “Parental Hesitancy on COVID-19 Vaccination of Children Under the Age of 16: A Cross-Sectional Mixed-Methods Study Among Factory Workers.” I very much enjoyed reading your manuscript.

Summary: This study investigates parental hesitancy toward COVID-19 vaccination for children under 16 among factory workers in Myanmar, utilizing a cross-sectional mixed-methods design. Quantitative data were gathered through structured interviews with 170 factory workers and analyzed using binary logistic regression to determine factors associated with vaccine hesitancy. The qualitative component involved in-depth interviews with 12 participants to explore reasons for hesitancy and non-hesitancy. The results indicate that 18.2% of parents were hesitant, 25.9% were unsure, and 55.9% were non-hesitant. Key factors associated with hesitancy included male gender (AOR: 3.04, 95% CI: 1.35–6.84) and lack of prior SARS-CoV-2 infection (AOR: 2.66, 95% CI: 1.06–6.70). Common concerns included child age, vaccine safety, uncertainty about effectiveness, and distrust in vaccine origin. The study suggests that health education programs and media campaigns should be strengthened to address misinformation and improve vaccination rates. While the study provides valuable insights into vaccine hesitancy within a specific workforce, further multi-center studies with a larger sample size and comparative analysis across different occupational groups could enhance its generalizability.

Major Comments:

1. Literature Review:

Comment 1: You may expand the discussion on global vaccine hesitancy trends, particularly in occupational settings, to contextualize the study findings.

Response: Thanks for pointing this out and we revised as following:

“Factory workers are essential for driving the production growth and play a crucial role in generating social wealth [16]. The effective COVID-19 vaccination program with the strategies of sustainable vaccination process among factory workers is vital for the recovery of economic sector [17]. Regarding the occupational status, frontline

---

## [Decision Letter · Decision Letter 1]

PONE-D-25-10068R1Parental hesitancy on COVID-19 vaccination of children under the age of 16: A cross-sectional mixed-methods study among factory workersPLOS ONE

Dear Dr. Htun,

Thank you for submitting your manuscript to PLOS ONE. After careful consideration, we feel that it has merit but does not fully meet PLOS ONE’s publication criteria as it currently stands. Therefore, we invite you to submit a revised version of the manuscript that addresses the points raised during the review process.

We look forward to receiving your revised manuscript.

Kind regards,

Harapan Harapan, MD, PhD

Academic Editor

PLOS ONE

Journal Requirements:

Reviewers' comments:

Reviewer's Responses to Questions

**Comments to the Author**

1. If the authors have adequately addressed your comments raised in a previous round of review and you feel that this manuscript is now acceptable for publication, you may indicate that here to bypass the “Comments to the Author” section, enter your conflict of interest statement in the “Confidential to Editor” section, and submit your "Accept" recommendation.

Reviewer #1: All comments have been addressed

Reviewer #3: All comments have been addressed

2. Is the manuscript technically sound, and do the data support the conclusions?

Reviewer #1: Yes

Reviewer #3: Yes

3. Has the statistical analysis been performed appropriately and rigorously? 

Reviewer #1: Yes

Reviewer #3: Yes

4. Have the authors made all data underlying the findings in their manuscript fully available?

Reviewer #1: No

Reviewer #3: Yes

5. Is the manuscript presented in an intelligible fashion and written in standard English?

Reviewer #1: Yes

Reviewer #3: Yes

6. Review Comments to the Author

Reviewer #1: The authours have apprporiately responded to almost all cmoments by the reviewers. However , in line 251 of the revised manuscrpit they refer to Figure 1-Flow chart of participant selection and data collection process .Similarly in line 322 they also refer to Figure 2-main sources of information on COVID--19 Vaccine among participants. The two figures were not included in the manuscript. These had been recommended by one of the reviewers, they should be included.

Reviewer #3: Thank you for giving me the opportunity to review your manuscript entitled “Parental hesitancy on COVID-19 vaccination of children under the age of 16: A cross-sectional mixed-methods study among factory workers” I very much enjoyed reading your manuscript.

Comments:

Thanks for addressing the previous reviewer comments. I have no further comments.

7. PLOS authors have the option to publish the peer review history of their article (what does this mean? ). If published, this will include your full peer review and any attached files.

**Do you want your identity to be public for this peer review?** For information about this choice, including consent withdrawal, please see our Privacy Policy .

Reviewer #1: **Yes: ** Edison Mworozi Arwanire

Reviewer #3: **Yes: ** Ibrahim Saleh

---

## [Author Response · Author response to Decision Letter 2]

17 May 2025

Responses to editor and reviewers

PONE-D-25-10068R1

Parental hesitancy on COVID-19 vaccination of children under the age of 16: A cross-sectional mixed-methods study among factory workers

PLOS ONE

Revised manuscript on May 18, 2025

Dear Prof. Harapan Harapan,

I would like to thank you again for the opportunity to revise and resubmit our manuscript titled “Parental hesitancy on COVID-19 vaccination of children under the age of 16: A cross-sectional mixed-methods study among factory workers”. We are grateful to you and the reviewers for the helpful comments on our manuscript. It is our belief that the manuscript is substantially improved after making the suggested edits.

Followings this letter are the editor and reviewers’ comments with our responses in point by point, including how and where the text was modified. Changes made in the revised manuscript are marked using track changes. The revision has been developed in consultation with all authors, and each author has given approval to the final form of this revision.

Thank you again for consideration of our revised manuscript.

Sincerely,

Ye Minn Htun

Health and Disease Control Unit

Nay Pyi Taw 15011, Myanmar

Journal Requirements

Response: Thanks for this comment. We reviewed and rechecked the references for ensuring that it is correct.

Reviewers' comments:

Review Comments to the Author

Reviewer #1:

Comment: The authours have apprporiately responded to almost all cmoments by the reviewers. However, in line 251 of the revised manuscrpit they refer to Figure 1-Flow chart of participant selection and data collection process. Similarly in line 322 they also refer to Figure 2-main sources of information on COVID--19 Vaccine among participants. The two figures were not included in the manuscript. These had been recommended by one of the reviewers, they should be included.

Response: Thanks for this comment. We prepared and submitted the figures with the separate files, according the submission guideline of PLOS One described as “Do not include figures in the main manuscript file. Each figure must be prepared and submitted as an individual file. Cite figures in ascending numeric order at first appearance in the manuscript file.” (https://journals.plos.org/plosone/s/submission-guidelines#loc-figures-and-tables)

(https://journals.plos.org/plosone/s/figures#loc-how-to-submit-figures-and-captions).

However, we followed the reviewers’ comment and added in the main manuscript (end of the manuscript).

Reviewer #3:

Comment: Thank you for giving me the opportunity to review your manuscript entitled “Parental hesitancy on COVID-19 vaccination of children under the age of 16: A cross-sectional mixed-methods study among factory workers” I very much enjoyed reading your manuscript.

Response: Thanks for your comments on our manuscript.

Comments: Thanks for addressing the previous reviewer comments. I have no further comments.

Response: Thanks for your comments and precious time.

---

## [Decision Letter · Decision Letter 2]

Parental hesitancy on COVID-19 vaccination of children under the age of 16: A cross-sectional mixed-methods study among factory workers

PONE-D-25-10068R2

Dear Dr. Htun,

We’re pleased to inform you that your manuscript has been judged scientifically suitable for publication and will be formally accepted for publication once it meets all outstanding technical requirements.

Kind regards,

Harapan Harapan, MD, PhD

Academic Editor

PLOS ONE

Additional Editor Comments (optional):

Reviewers' comments:

Reviewer's Responses to Questions

**Comments to the Author**

1. If the authors have adequately addressed your comments raised in a previous round of review and you feel that this manuscript is now acceptable for publication, you may indicate that here to bypass the “Comments to the Author” section, enter your conflict of interest statement in the “Confidential to Editor” section, and submit your "Accept" recommendation.

Reviewer #1: All comments have been addressed

Reviewer #3: All comments have been addressed

2. Is the manuscript technically sound, and do the data support the conclusions?

Reviewer #1: Yes

Reviewer #3: Yes

3. Has the statistical analysis been performed appropriately and rigorously? 

Reviewer #1: Yes

Reviewer #3: Yes

4. Have the authors made all data underlying the findings in their manuscript fully available?

Reviewer #1: Yes

Reviewer #3: Yes

5. Is the manuscript presented in an intelligible fashion and written in standard English?

Reviewer #1: Yes

Reviewer #3: Yes

6. Review Comments to the Author

Reviewer #1: The Authours are commended for having fully responded to the comments/recommdations by the reviewers

Reviewer #3: Thank you for your thoughtful and detailed responses to the reviewers’ comments. I have no additional comments.

7. PLOS authors have the option to publish the peer review history of their article (what does this mean? ). If published, this will include your full peer review and any attached files.

**Do you want your identity to be public for this peer review?** For information about this choice, including consent withdrawal, please see our Privacy Policy .

Reviewer #1: **Yes: ** Edison Mworozi Arwanire

Reviewer #3: **Yes: ** Ibrahim Saleh

---

## [Editor Report · Acceptance letter]

PONE-D-25-10068R2

PLOS ONE

Dear Dr. Htun,

I'm pleased to inform you that your manuscript has been deemed suitable for publication in PLOS ONE. Congratulations! Your manuscript is now being handed over to our production team.

Kind regards,

on behalf of

Dr. Harapan Harapan

Academic Editor

PLOS ONE